# Impact of social restrictions during the COVID-19 pandemic on the physical activity levels of adults aged 50–92 years: a baseline survey of the CHARIOT COVID-19 Rapid Response prospective cohort study

David Salman ![ORCID] ,[1,2] Thomas Beaney,[1] Catherine E Robb,[3] Celeste A de Jager Loots ![ORCID] ,[3] Parthenia Giannakopoulou,[3] Chinedu T Udeh-Momoh,[3] Sara Ahmadi-Abhari,[3] Azeem Majeed,[1,4] Lefkos T Middleton,[3,4] Alison H McGregor[2]

DS, TB and CER are joint senior authors.

For numbered affiliations see end of article.

**Correspondence to**
Dr David Salman;
d.salman11@imperial.ac.uk and
Dr Thomas Beaney;
thomas.beaney@imperial.ac.uk

## ABSTRACT

**Objectives** Physical inactivity is more common in older adults, is associated with social isolation and loneliness and contributes to increased morbidity and mortality. We examined the effect of social restrictions to reduce COVID-19 transmission in the UK (lockdown), on physical activity (PA) levels of older adults and the social predictors of any change.

**Design** Baseline analysis of a survey-based prospective cohort study.

**Setting** Adults enrolled in the Cognitive Health in Ageing Register for Investigational and Observational Trials cohort from general practitioner practices in North West London were invited to participate from April to July 2020.

**Participants** 6219 cognitively healthy adults aged 50–92 years completed the survey.

**Main outcome measures** Self-reported PA before and after the introduction of lockdown, as measured by metabolic equivalent of task (MET) minutes. Associations of PA with demographic, lifestyle and social factors, mood and frailty.

**Results** Mean PA was significantly lower following the introduction of lockdown from 3519 to 3185 MET min/week (p<0.001). After adjustment for confounders and prelockdown PA, lower levels of PA after the introduction of lockdown were found in those who were over 85 years old (640 (95% CI 246 to 1034) MET min/week less); were divorced or single (240 (95% CI 120 to 360) MET min/week less); living alone (277 (95% CI 152 to 402) MET min/week less); reported feeling lonely often (306 (95% CI 60 to 552) MET min/week less); and showed symptoms of depression (1007 (95% CI 612 to 1401) MET min/week less) compared with those aged 50–64 years, married, cohabiting and not reporting loneliness or depression, respectively.

**Conclusions and implications** Markers of social isolation, loneliness and depression were associated with lower PA following the introduction of lockdown in the

## Strengths and limitations of this study

► Out of 40 000 people contacted, 7320 responded and 6219 completed the survey.
► A significant reduction in mean levels of physical activity (PA) was found in older adults after the introduction of lockdown measures.
► Multivariable analyses were adjusted for confounders according to predetermined causal pathways.
► Survey responders identified predominantly as White/Caucasian background, and showed higher levels of PA than the general population, which may limit the generalisability of the findings to other population groups.
► The potential for recall bias from using a self-report questionnaire for PA levels (International Physical Activity Questionnaire). This includes reliance on recall for prelockdown PA levels.

UK. Targeted interventions to increase PA in these groups should be considered.

## BACKGROUND AND RATIONALE

Physical inactivity adversely affects older adults, with more than 60% of those aged over 75 years not sufficiently physically active for good health[1 2] as defined by meeting the WHO[3] and UK[4] guidelines. From March until June 2020 in the UK, a national 'lockdown' was implemented to reduce exposure to, and transmission of, COVID-19. Although applied to the whole population, adults aged over 70 years and those with underlying health conditions at higher risk of severe COVID-19 disease were asked to follow more stringent social distancing measures. These included

remaining at home where possible; avoiding social mixing in the community; avoiding physically interacting with friends and family; and avoiding public transport (online supplemental figure S1).[5]

Social isolation and loneliness in older adults, possibly exacerbated during lockdowns,[6] are associated with increases in morbidity and mortality, increased physical inactivity and sedentary time[7 8] and reduced physical performance.[9] Physical inactivity may therefore have a role in mediating the increased morbidity and mortality associated with social isolation.[10] Physical activity (PA) is important in the prevention of sarcopenia, frailty and decreased functional ability in older adults.[11] Data collected on the pandemic, predominantly in younger adults and children, suggest a decrease in PA and an increase in sedentary time.[12] Given the increased susceptibility to physical inactivity and social isolation in older adults in particular, this is an important area of study.[13] We set up the Cognitive Health in Ageing Register for Investigational and Observational Trials (CHARIOT) COVID-19 Rapid Response (CCRR) study in April 2020 to monitor the symptoms and the impact of the COVID-19 pandemic on various health and lifestyle factors by repeat questionnaire survey of the CHARIOT members.

We hypothesised that imposed social restrictions would negatively impact on PA levels of older adults, and that change in PA after the introduction of lockdown would be modified by certain demographic, lifestyle and social factors, with a focus on markers of social isolation and perceived loneliness. An awareness of the extent of, and predictors for, change in PA levels will aid our understanding of the impact of social isolation on the health of older adults, both with respect to pandemic-related lockdowns and social isolation itself.

## METHODS
### CCRR survey
Study participants were recruited from the CHARIOT register, a cohort of over 40 000 cognitively healthy (without a known diagnosis of dementia) adult volunteers aged over 50 years, recruited from 172 general practitioner (GP) surgeries across West and North London as part of a collaboration between regional GP practices and the School of Public Health at Imperial College London.

This ongoing prospective cohort study was initiated in April 2020 with repeated questionnaire surveys conducted every 6 weeks. The CCRR baseline survey consists of questions related to basic demographics, diet, alcohol and smoking status, symptoms of COVID-19, functional activities, PA, sleep, frailty and mental health (online supplemental file 2). For PA, the International Physical Activity Questionnaire (IPAQ) Short Form (last 7 days) was used,[14] asking respondents to document their weekly vigorous and moderate activities, walking and sitting time from the week prior to completing the survey; and for the week prior to implementation of social restriction measures. This has test–retest reliability of 0.75

in those under the age of 60 years.[15] However, although less commonly studied in older populations, one study demonstrated reduced reliability at 0.65 and 0.57 for men and women, respectively, aged 65–74 years, and 0.50 and 0.56 for those aged 75–89 years, but with adequate validity when assessed against objective measures.[16] For assessing frailty, the 5-point FRAIL (Fatigue, Resistance, Aerobic, Illnesses, Loss of weight) scale[17 18] (ordinal scale 1–5; predictive validity for mortality up to 10 years; HR 2.60)[19] and for assessing mental health symptoms, the Hospital Anxiety and Depression Scale (HADS)[20]; sensitivity and specificity of 0.8 for both anxiety and depression[21]; 14 questions on feelings related to anxiety and depression rated on a 4-level Likert scale were used. A question on loneliness was used from the Imperial College Sleep Quality Questionnaire; in turn adapted from the Pittsburgh Sleep Quality Index[22] and Centre for Epidemiologic Studies of Depression Scale[23] for work-free periods.

Participants were eligible for recruitment if they were participating in the CHARIOT register, or were a consenting member of the household of a participant who wished to take part; had mental capacity to consent to participate; were willing and able to undertake an electronic questionnaire survey; were able to read, write and were fluent in English, or identify an informant who was. Participants were excluded where they were no longer participating in the CHARIOT register, or if they did not have access to electronic devices to complete the questionnaire surveys. Survey data used in the present analysis were completed between 30 April and 22 July 2020, and a timeline of lockdown measures has been incorporated into the online supplemental figure S1.

### Statistical analysis
All analyses were conducted using Stata V.16.1 (StataCorp, 2019) and R.[24 25] Body mass index (BMI) was calculated as weight in kilograms divided by the square of height in metres and categorised according to standard WHO criteria. IPAQ data were cleaned according to the IPAQ data cleaning protocol,[26] and the metabolic equivalent of task (MET) min/week, calculated for each activity and total activity (where 3.3 METs is considered equivalent to walking, and moderate and vigorous activities equivalent to 4 and 8 METs, respectively). Periods of activity under 10 min were excluded as per the protocol, excluding for vigorous, moderate and walking activities during lockdown 25, 23 and 12 periods, respectively (for prelockdown activity, excluding 10, 13 and 3 periods of activity, respectively). To calculate the total MET min/week, the self-reported duration (minutes) and frequency (days) of each of these PA categories is multiplied by the specified metric (online supplemental methods). Paired t-tests were used to compare the distributions of mean PA levels before and following the introduction of lockdown.

Measures of association with explanatory variables were explored in univariable linear regression models for two outcomes: (1) overall weekly MET minutes after introduction of lockdown and (2) the difference in overall weekly

MET minutes before versus after the introduction of lockdown. Multivariable models were constructed for the outcome of MET minutes after the introduction of lockdown, adjusting a priori each explanatory variable in turn for age, sex and ethnicity. Month of survey completion was also included in the model to account for seasonal changes, and the finding that PA after the introduction of lockdown varied by month (online supplemental figure S2 and table S1). Weekly MET minutes before the introduction of lockdown were also included in the model given its strong association with activity levels after the introduction of lockdown, which remained significantly associated in all models. Denominators for each model vary according to the levels of missingness in variables included in the models, which was low for most variables, except for BMI (unrecorded in 51.4% of participants). Employment was recategorised into four groups for the purposes of regression analysis (online supplemental table S2).

A causal diagram was constructed using DAGitty[27] (online supplemental figure S3) to aid adjustment for confounders in order to separate the overall causal effects of marital status, loneliness and living alone on PA. Additional multivariable models were then constructed based on the causal diagram for loneliness, adjusting for age, sex, ethnicity, household status, marital status, shielding status and frailty category. No further adjustment was necessary for marital status or household status. Residuals were plotted against fitted values to assess for outlying points and heteroscedasticity; and plots of Cook's distance and leverage against fitted values were examined to detect the presence of influential points.

### Patient and public involvement
Older adult volunteers (60–80 years of age) from various social and cultural backgrounds provided feedback on the survey content. This feedback was incorporated into the survey design. Participants in the CHARIOT cohort are informed by regular newsletter of all publications pertaining to the cohort.

### RESULTS
#### Participant characteristics
The survey was sent to 15 000 CHARIOT participants via email, with a subsequent 25 000 contacted by post. A total of 7320 participants responded and completed the survey. Of these respondents, 6219 completed IPAQ data both before and after introduction of lockdown measures and were included in the final analysis.

Of the 6219 participants included in the present study, 55.4% were female, and the majority (55.3%) were aged 65–74 years with a mean age of 70 years. 93.7% of respondents classified themselves as being of White ethnic background, with 2.8% of Asian ethnic background, and only 0.7% of Black African or Caribbean background. Approximately half of participants (48.6%) had a recorded height and weight, with a mean BMI of 25.3 kg/m$^2$. The

majority of respondents were married (62.2%), cohabiting (72.8%) and retired (69.5%). Most respondents did not smoke (96.9%), drank alcohol (82.6%) and felt they ate a healthy diet (80.3%). 18.0% of respondents were classified as prefrail, with 0.5% as frail and 26.2% reported that they were shielding at the time of the survey (table 1).

### PA before and after social distancing measures
Mean (SD) PA for participants prior to lockdown was 3519 (2867) MET min/week. There was a significant reduction in mean MET minutes following implementation of lockdown to 3185 (2673) MET min/week (p<0.001; table 2). A total of 3167 (50.9%) participants decreased their activity following the introduction of lockdown by a mean (SD) of 1957 (2025) MET min/week, 534 (8.6%) maintained the same level of activity and 2518 (40.5%) increased their activity by a mean (SD) of 1636 (1775) MET min/week. Mean sitting time increased by 276 MET min/week after the introduction of lockdown (2680) compared with before (2404) (table 2).

A total of 5762 (92.7%) participants achieved at least the minimum guidance of 600 MET min/week of activity, as defined by WHO,[3] prior to implementation of lockdown measures, slightly reducing to 5672 (91.2%) following their introduction (p<0.001). A total of 5039 (81.0%) achieved 1200 MET min/week before lockdown, with 4904 (78.9%) achieving this after the introduction of lockdown (p<0.001, online supplemental figure S4). Following the introduction of lockdown, PA levels varied by month of survey completion, with the highest levels in June and lowest levels in July. There was no significant difference between self-reported PA before lockdown by month of survey completion (online supplemental figure S5).

### Predictors of PA after the introduction of lockdown and change from before lockdown
#### Demographic and lifestyle factors
Univariable linear regression models (online supplemental table S3) showed statistically significant associations with lower PA after the introduction of lockdown in older age groups (p<0.001; figure 1), but no evidence of differences in the change from before lockdown between age groups (p=0.184; figure 2). After multivariable adjustment for age, sex, ethnicity, month of survey completion and prelockdown PA (online supplemental table S4), there was evidence of significantly lower levels of PA with increasing age, with adults aged 85 years and over doing on average 640 (95% CI 246 to 1034) MET min/week less than those aged 50–64 years (figure 3). There was no significant difference in PA after the introduction of lockdown in males and females (p=0.180; figure 1), but females on average exhibited a greater decline in PA from before lockdown than males (450 vs 189 MET min/week less respectively; p<0.001; figure 2). After multivariable adjustment, there was only a small and borderline significant difference in PA after lockdown was introduced

**Table 1** Participant characteristics for 6219 participants with complete data on physical activity

| Participant characteristic | | Total | % |
|---|---|---|---|
| Gender | Female | 3445 | 55.4 |
| | Male | 2770 | 44.5 |
| | Prefer not to say | 4 | 0.1 |
| | Mean (SD) | 69.9 (7.3) | |
| | Median (IQR) | 70 (66–74) | |
| | Range | 50–92 | |
| Age (years) | 50–64 | 1212 | 19.5 |
| | 65–74 | 3440 | 55.3 |
| | 75–84 | 1394 | 22.4 |
| | 85+ | 127 | 2.0 |
| | Missing data | 46 | 0.7 |
| Ethnicity | White | 5825 | 93.7 |
| | English/Welsh/Scottish/Northern Irish/British | 5143 | 82.7 |
| | Any other White background | 536 | 8.6 |
| | Irish | 146 | 2.3 |
| | Mixed/multiple ethnic groups | 99 | 1.6 |
| | White and Black African | 10 | 0.2 |
| | White and Asian | 33 | 0.5 |
| | White and Black Caribbean | 7 | 0.1 |
| | Any other mixed/multiple ethnic background | 49 | 0.8 |
| | Asian/Asian British | 174 | 2.8 |
| | Indian | 91 | 1.5 |
| | Pakistani | 12 | 0.2 |
| | Bangladeshi | 2 | 0.0 |
| | Chinese | 32 | 0.5 |
| | Any other Asian background | 37 | 0.6 |
| | Black/African/Caribbean/Black British | 43 | 0.7 |
| | African | 13 | 0.2 |
| | Caribbean | 21 | 0.3 |
| | Any other Black/African/Caribbean/Black British | 9 | 0.1 |
| | Other ethnic group | 64 | 1.0 |
| | Arab | 17 | 0.3 |
| | Any other ethnic group | 47 | 0.8 |
| | Prefer not to say | 14 | 0.2 |
| | Mean (SD) | 25.3 (5.1) | |
| | Median (IQR) | 24.4 (22.2–27.1) | |
| Body mass index (BMI) (kg/m$^2$) | <18.5 (underweight range) | 61 | 1.0 |
| | 18.5–24.9 (healthy weight) | 1644 | 26.4 |
| | 25.0–29.9 (overweight) | 962 | 15.5 |
| | ≥30.0 (obese range) | 358 | 5.8 |
| | Missing data | 3194 | 51.4 |
| Shielding at time of questionnaire | No | 4591 | 73.8 |
| | Yes | 1628 | 26.2 |
| Marital status | Married | 3869 | 62.2 |
| | Single | 789 | 12.7 |
| | Widowed | 601 | 9.7 |
| | Divorced | 595 | 9.6 |

Continued

**Table 1** Continued

| Participant characteristic | | Total | % |
|---|---|---|---|
| Living arrangements | Living with a partner | 365 | 5.9 |
| | Cohabiting | 4530 | 72.8 |
| | Living alone | 1689 | 27.2 |
| Employment | Retired | 4322 | 69.5 |
| | Continuing to work in your usual job; at home | 1101 | 17.7 |
| | None of the above | 201 | 3.2 |
| | Furloughed (put on leave, still getting paid) | 197 | 3.2 |
| | Continuing to work in your usual job and leave home for your job | 141 | 2.3 |
| | A key worker | 96 | 1.5 |
| | Had to close your business due to COVID-19 | 70 | 1.1 |
| | Lost my job due to the lockdown | 42 | 0.7 |
| | Unemployed | 36 | 0.6 |
| | A student | 13 | 0.2 |
| Current smoker | No | 6027 | 96.9 |
| | Yes | 192 | 3.1 |
| Alcohol intake | No | 1083 | 17.4 |
| | Yes | 5136 | 82.6 |
| Diet | No change from usual—already had a healthy diet. | 4991 | 80.3 |
| | My diet has become more healthy. | 715 | 11.5 |
| | My diet was healthy before but has got worse since lockdown. | 312 | 5.0 |
| | No change from usual—my diet is not very healthy. | 201 | 3.2 |
| Frailty | Robust | 5055 | 81.3 |
| | Prefrail | 1117 | 18.0 |
| | Frail | 34 | 0.5 |
| | Missing data | 13 | 0.2 |
| Loneliness | Not ever | 2994 | 48.1 |
| | Rarely | 1469 | 23.6 |
| | Sometimes | 1305 | 21.0 |
| | Often | 372 | 6.0 |
| | Missing data | 79 | 1.3 |
| HADS (depression score) | Normal (0–7) | 4658 | 74.9 |
| | Borderline (8–10) | 312 | 5.0 |
| | Abnormal (11–21) | 116 | 1.9 |
| | Missing data | 1133 | 18.2 |
| HADS (anxiety score) | Normal (0–7) | 4335 | 69.7 |
| | Borderline (8–10) | 486 | 7.8 |
| | Abnormal (11–21) | 265 | 4.3 |
| | Missing data | 1133 | 18.2 |
| Total participants | | 6219 | |

HADS, Hospital Anxiety and Depression Scale.

between gender (PA in males on average 108 MET min/week more than females; 95% CI −1 to 216; figure 3). No significant associations were seen between PA after the introduction of lockdown or change in PA according to ethnicity or employment status before or after adjustment.

Lower levels of PA after the introduction of lockdown were seen with increasing BMI category in current smokers and in those reporting an unhealthy or worsening diet before and after adjustment (figure 1). After adjustment, a dose–response relationship was evident between lower PA and increasing BMI (p=0.030), with obese individuals doing 578 (95% CI 324 to 832) MET min/week less than those of a healthy weight (figure 3). The denominator included in analyses of BMI was significantly lower than for other models, as BMI was unrecorded for 51.4% of participants. Current alcohol consumption was weakly

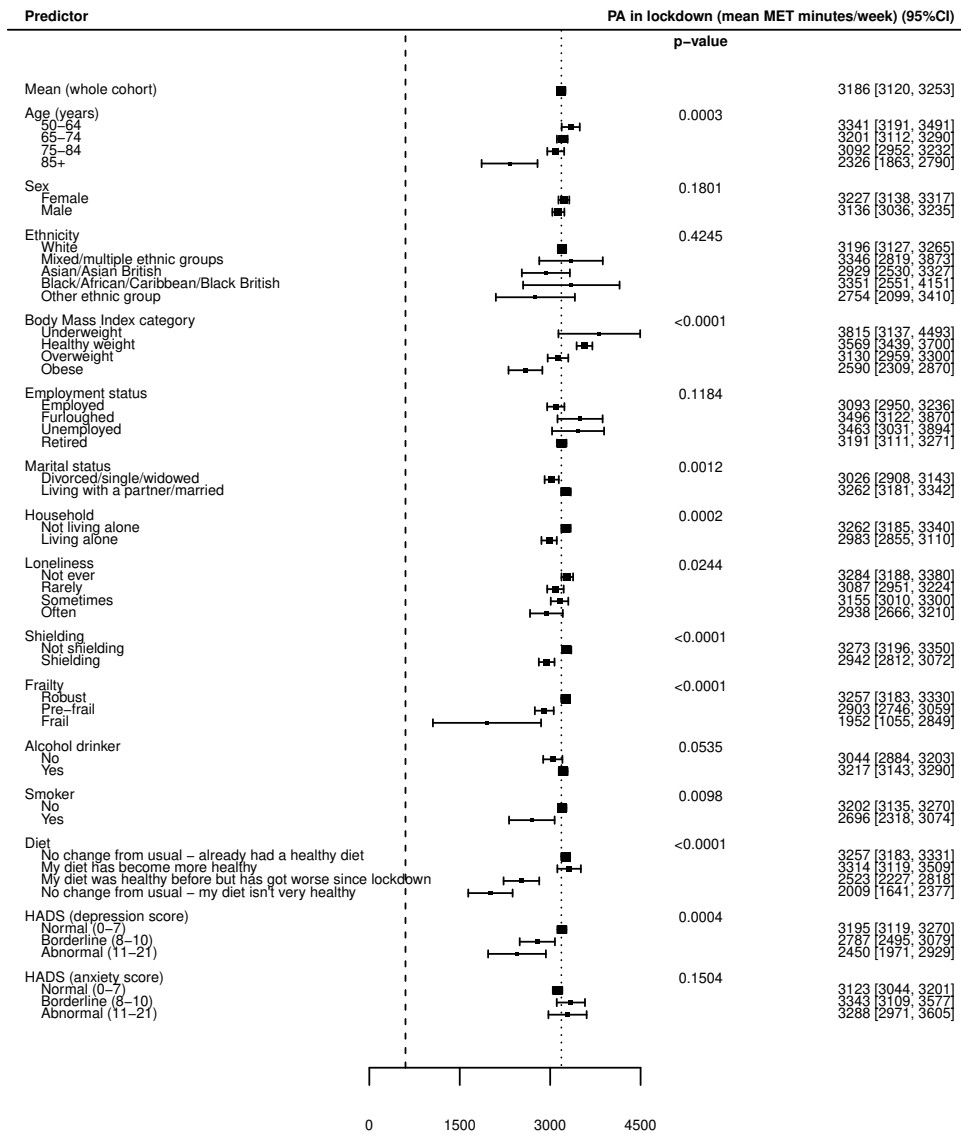

**Figure 1** Forest plot of unadjusted univariable associations with physical activity (PA) following the introduction of lockdown measures (during/in lockdown). Data presented as mean MET min/week±95% CI. Heavy dashed line—600 MET min/week (WHO minimal PA guideline for adults); light dashed line—mean MET minutes for the whole cohort. See also online supplemental table S3. HADS, Hospital Anxiety and Depression Scale; MET, metabolic equivalent of task.

associated with increased levels of PA in both univariable and multivariable models, with current drinkers reporting 145 MET min/week more than non-drinkers after adjustment (95% CI 1 to 289; figures 2 and 3).

### Associations with social isolation and loneliness

Participants who were divorced, single or widowed were, on average, less active after the introduction of lockdown than those married or living with a partner (3026 vs 3262 MET min/week; p=0.001; figure 1), and exhibited a greater decline in PA from before lockdown (540 vs 236 MET min/week less; p<0.001; figure 2). The association with PA after the introduction of lockdown remained after adjustment, with those divorced, single or widowed doing on average 240 (95% CI 120 to 360) MET min/

week less (figure 3). Participants living alone were also less active than those cohabiting and showed greater reductions in PA from before lockdown. After adjustment for confounders and PA before lockdown, those living alone were doing 277 (95% CI 152 to 402) MET min/week less than those cohabiting (figure 3).

Significant associations were seen between PA after the introduction of lockdown and frequency of loneliness, with those 'often' experiencing loneliness achieving 2938 MET min/week compared with 3284 MET min/week in those 'never' experiencing loneliness (p=0.024; figure 1). Greater declines in PA from before lockdown were also seen with increasing loneliness (figure 2). After adjustment, PA after the introduction of lockdown was

**Table 2** Physical activity and sitting time for recipients before and following introduction of lockdown measures

| Physical activity type | | Before | During | P value for difference |
|---|---|---|---|---|
| Vigorous activity | Mean (SD) min/week | 145 (276) | 135 (253) | 0.004 |
| | Median (IQR) min/week | 40 (0–180) | 10 (0–180) | |
| Moderate activity (min/week) | Mean (SD) min/week | 292 (430) | 245 (374) | <0.001 |
| | Median (IQR) min/week | 120 (0–360) | 120 (0–360) | |
| Walking (min/week) | Mean (SD) min/week | 462 (460) | 403 (408) | <0.001 |
| | Median (IQR) min/week | 360 (150–630) | 315 (150–525) | |
| Sitting (min/week)* | Mean (SD) min/week | 2404 (1137) | 2680 (1181) | <0.001 |
| | Median (IQR) min/week | 2100 (1680–2940) | 2520 (1680–3360) | |
| MET min/week | Mean (SD) min/week | 3519 (2867) | 3185 (2673) | <0.001 |
| | Median (IQR) min/week | 2772 (1386–4650) | 2440 (1386–4185) | |

Data presented as min/week with both mean (SD) and median (IQR) shown. pvalues from paired t-test.
*Denominator 6023.
MET, metabolic equivalent of task.

significantly lower for those with increased frequency of loneliness (figure 3). After full adjustment including, in addition, household status, marital status, shielding status and frailty category, those experiencing loneliness 'often' reported 306 (95% CI 60 to 552) MET min/week less activity than those 'never' lonely (online supplemental table S5).

Significantly lower PA levels were recorded in those shielding and in participants categorised as prefrail or frail (both p<0.001; figure 1). Larger declines in PA from before lockdown were seen in those shielding compared with those not shielding (588 vs 243 MET min/week less; p<0.001; figure 2), but there was no significant difference in change in PA according to frailty category (p=0.389; figure 2). After adjustment, frail participants were doing 926 (95% CI 189 to 1663) MET min less on average than those classed as robust (figure 3). Participants who were shielding were doing an average of 290 (95% CI 163 to 417) MET min/week less than those not shielding (figure 3).

### Associations with depression and anxiety

Symptoms of depression were associated with lower levels of PA following the introduction of lockdown, with those meeting the criteria for depression reporting 2450 MET min/week compared with 3195 MET min/week in those with normal scores (p<0.001; figure 1). There was no strong association with anxiety scores. Mean change in PA from before lockdown was associated with both depression and, in contrast to absolute PA levels, with anxiety scores. Participants with depression reported 1450 MET min/week less on average after lockdown was introduced compared with before, while those with normal scores reported 293 MET min/week less (p<0.001; figure 2). Similarly, in those with anxiety, PA reduced by 836 MET min/week compared with 312 MET min/week in those with normal scores (p=0.004; figure 2).

After adjustment, those meeting the criteria for depression on the HADS scale had significantly lower PA levels than those with normal scores, doing on average 1007

(95% CI 612 to 1401) MET min/week less (figure 3). There remained no statistically significant association between anxiety score and PA after adjustment.

## DISCUSSION
### Main findings

Data from the CCRR study show that participants experienced, on average, a significant decrease in PA after the introduction of lockdown in the UK when compared with before, together with an increase in sitting time. When adjusted for age, sex, ethnicity, month of survey completion and baseline PA, factors strongly associated with a reduction in PA include: increased age, increased BMI, frailty, current smoking and a change to a less healthy diet. Factors associated with social isolation were also significantly associated with a reduction in PA: those divorced, single or widowed, living alone, shielding or reporting increased frequency of loneliness did significantly less PA after lockdown was introduced. Furthermore, a strong association was also seen with lower PA following the introduction of lockdown in those with depression, but not for those with anxiety.

### The effect of lockdown on PA

There was a reduction in PA in over half of our participants, and a decrease in mean levels of PA by 333 MET min/week following the introduction of lockdown measures in the UK. This was accompanied by an increase in sitting time by 276 min/week, an adverse finding given the adverse health impacts associated with increased sedentary and sitting time.[28] These findings correlate with other studies from the UK (a decrease in 25% of adults aged over 20 years following lockdown),[29] Spain[30] and China,[31] and from a global survey collected in eight different languages,[32] despite the differences in outdoor exercise permissions between countries. Reductions in PA may impact disproportionately across society. We found that increasing age associated with a reduction in PA after lockdown was introduced, corresponding with that seen

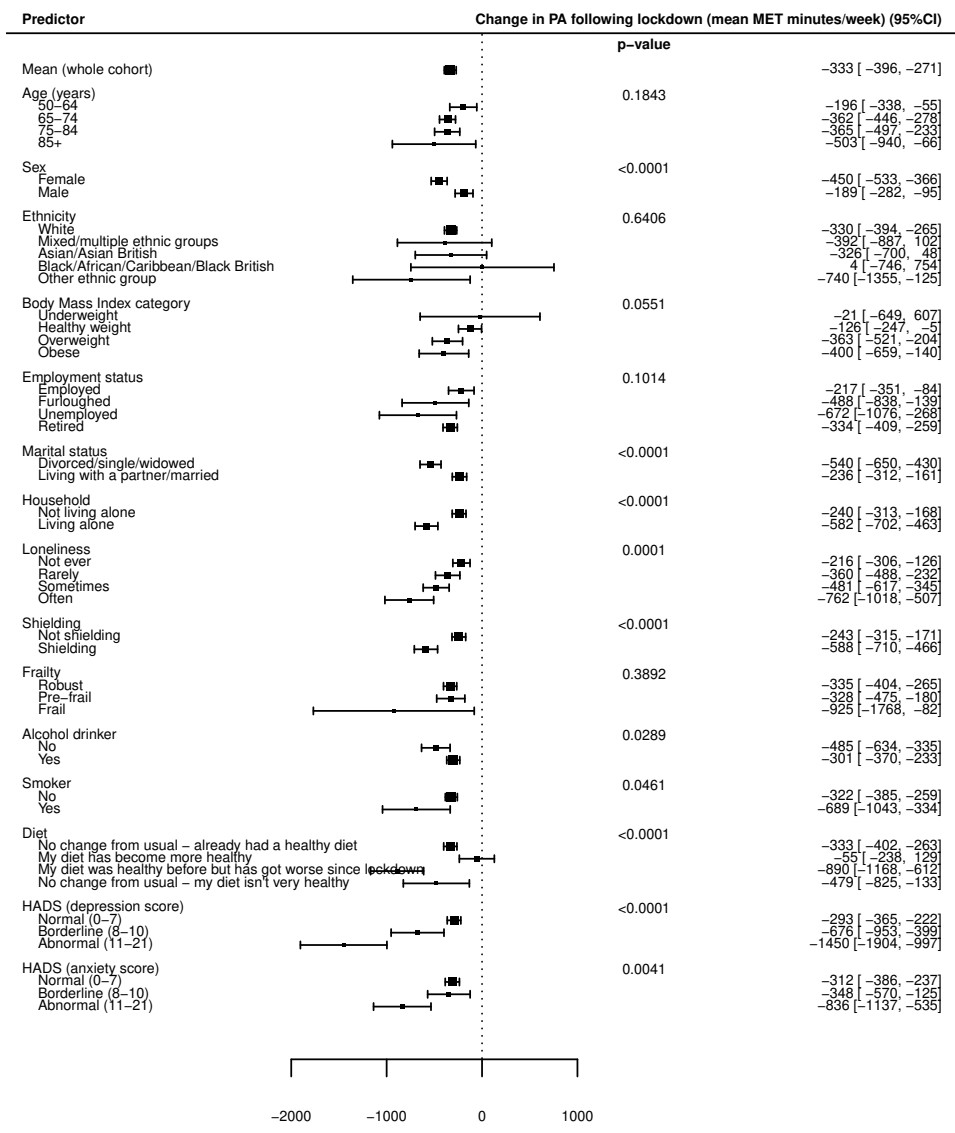

**Figure 2** Forest plot of unadjusted mean change in physical activity (PA), following the introduction of lockdown from before, for all variables (mean MET min/week±95% CI). Negative values indicate a decline in activity after the introduction of lockdown when compared with before. See also online supplemental table S3. HADS, Hospital Anxiety and Depression Scale; MET, metabolic equivalent of task.

in Japan, with a 26.5% (65 min) decrease in total PA in adults aged 65–84.[33] The UK Active Lives Survey found a 7.3% reduction in the proportion of active adults aged 55–74 years from 63% to 56% during the pandemic, and a 6.6% reduction in those aged 75 years and above from 42% to 35%.[34] A study of self-reported data in the UK found that those with a diagnosis of obesity, hypertension, lung disease, depression or a disability were more likely to reduce PA during lockdown.[29]

### Social relationships, loneliness and PA

Individuals for whom social engagement was more likely to be restricted, such as those who were shielding, divorced, single, widowed or living alone, were more likely to have lower levels of PA after lockdown implementation, and

to have declined to a greater extent. Similarly, those who subjectively reported feeling lonely were more likely to have lower PA levels and greater declines from before lockdown. These associations remained significant after multivariable adjustment.

Associations between health behaviours, including PA, and social relationships have been noted previously. Data from the English Longitudinal Study of Ageing (ELSA) showed that socially isolated respondents were less likely to report healthy diets and more likely to smoke.[7] Crucially, they showed reduced activity counts in socially isolated individuals (measured by accelerometer) in a sample of adults older than 50 years,[8] and reduced self-reported moderate to vigorous PA.[7]

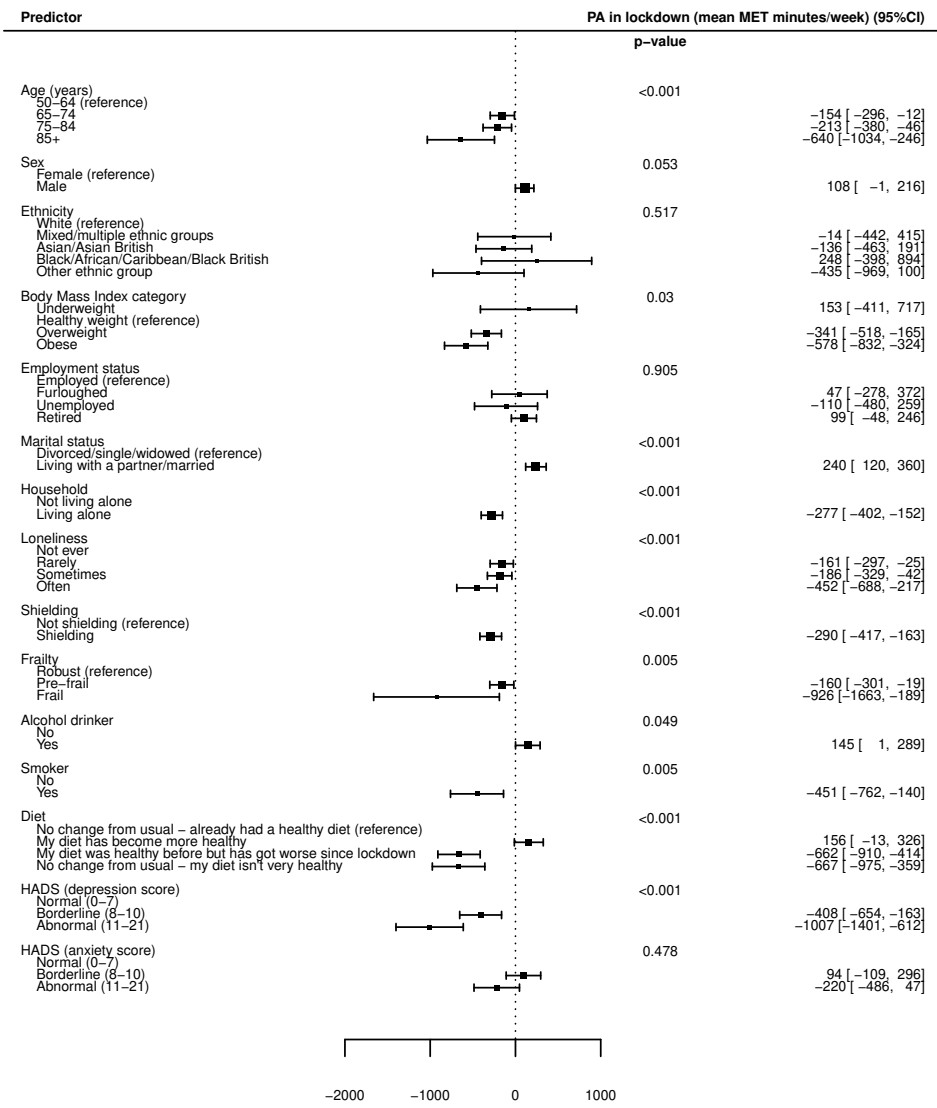

Multivariable associations with physical activity during lockdown (MET minutes/week)(95%CI)

**Figure 3** Forest plot of multivariable associations with physical activity (PA) after the introduction of lockdown (during/in lockdown), adjusted for age, sex, ethnicity, month of year of survey completion and baseline PA. Data presented as mean MET min/week±95% CI, compared with the reference group, with negative values indicating lower PA than the reference. See also online supplemental table S4. HADS, Hospital Anxiety and Depression Scale; MET, metabolic equivalent of task.

This is particularly important given that isolated and lonely individuals are at an increased risk of morbidity and mortality from cardiovascular events, with the majority of this association mediated by risk factors which include physical inactivity.[35] Fixed effect models from the ELSA cohort show that social disengagement, domestic isolation and loneliness are associated with measures of poorer physical performance and, although they appear to be independent of PA, may still be associated along the causal pathway.[9] Studies of spousal pairs found that both men and women in married couples had greater levels of PA than their single counterparts,[36] and changes in PA are positively associated with changes in the PA of a spouse.[37] Increasing PA is associated with larger,[38 39] more diverse[40] and more heterogenous (in

terms of PA) social networks, and having more physically active people in a social network is associated with being more active.[41]

The interaction between social relationships and PA levels may be bidirectional. Levels of PA are influenced by multiple factors at different levels, including individual (psychological, genetic); interpersonal (social networks); environmental (social, built, natural); and regional or global determinants.[42] Social networks might influence PA through social support for individuals to take up and maintain activity, but also by regulating social norms, and associating PA with social connections or attachments.[43] There may also be increased opportunities for PA[41] when social networks are present.

## Mood and PA

In those reporting symptoms of depression, there were significantly lower levels of PA and a significant decrease in activity when compared with before lockdown. These findings correlate with those from the UK,[44] Australia[45] and Spain,[46] which found inverse associations between PA levels and poor mental health. Similarly, a cross-sectional study of Brazilian adults who were self-isolating found lower odds of symptoms of anxiety or depression in those who were performing over 30 or 15 min/day of moderate or vigorous activity respectively, and higher odds in those with prolonged sedentary time over 10 hours.[47] The associations between PA and mental health are well known, with positive impacts on well-being,[48] and reduced incidence and severity of symptoms of mental ill health.[49–51] Therefore, these findings are unsurprising, although the interaction between PA and reduced markers of mental ill health in older adults may be bidirectional. Moreover, social isolation and loneliness may mediate some of this effect: previous data from the CCRR cohort showed an interaction between social isolation, loneliness and female gender with worsening depression and anxiety over lockdown.[52] We found no statistically significant difference in PA following the introduction of lockdown with anxiety symptoms, at odds with previous studies.[44] However, the trajectory of anxiety symptoms is not known, and it is not clear whether anxiety symptoms pre-dated the introduction of lockdown.

## Health behaviours and PA

A decrease in PA was associated with other detrimental health behaviours, including unhealthy diet and smoking. A similar tendency of clustering of unhealthy behaviours during the COVID-19 pandemic was noted in a cohort of patients with type 2 diabetes mellitus in Spain, who showed an increase in sugary foods and snack consumption alongside an increase in sitting time, and a decrease in time spent walking or doing moderate PA during lockdown when compared with beforehand.[53] That detrimental health behaviours might coincide in response to lockdown shows the importance of targeted interventions for certain groups. Interestingly, alcohol consumption was seen to be a protective factor in our cohort, and this does not tie with other findings on the negative associations with increased alcohol use during the COVID-19 pandemic.[54] This may be due to the specific demographic features of our cohort, but the possibility of alcohol consumption being associated with social interaction in this group cannot be excluded.

## Limitations

This study has several limitations which may impact the generalisability of our findings. First, the CCRR cohort appear more physically active than the general population. Ninety per cent of participants in CCRR achieved minimum WHO (2010)[3] guidance, both before and following the introduction of lockdown. Over 78% achieved double this amount, and mean levels of PA were at least five times greater than the minimum recommendation. In contrast, only 61% of UK adults aged 55–74 years achieve minimum recommended WHO (2010) levels.[2] Despite this, CCRR participants may still not be active enough for major health gains. A 2016 systematic review and meta-analysis suggested that optimal risk reduction for breast and colorectal cancer, diabetes, ischaemic heart disease and stroke events was obtained from PA at 3000–4000 MET min/week.[55]

Second, there are differences in demography between the CCRR cohort and the general population of the UK, which may explain the higher levels of PA we observed. Ninety-three per cent of CCRR respondents identify as White/Caucasian ethnicity. The Active Lives Survey demonstrated a difference in those achieving minimum activity levels in White British individuals (65%) and those from Black (58%) and Asian (54%) ethnicities.[2] Third, the CCRR survey relies on self-report using the IPAQ Short Form. IPAQ data are well validated across diverse participants up to the age of 65 years[14] and a study of the performance of the IPAQ in older Japanese adults demonstrated adequate validity.[16] However, results from self-reporting tools for PA only weakly correlate with those from objective measures, such as accelerometers and pedometers.[56–59] Finally, recall bias and seasonal changes in PA may also have impacted on the results, with the additional factor that data were collected remotely rather than face to face (although this was necessary due to pandemic control measures). The CCRR survey was collected in April to July 2020, with participants asked to recall PA levels in the week before lockdown, which over time may become less reliable. However, no significant differences were found in the mean PA levels reported before lockdown according to month of survey completion, and although there were apparent differences in PA following the introduction of lockdown by month, we were able to adjust for this in multivariable models. Furthermore, social restriction measures are dynamic and change over time, with a loosening of restrictions by 4 July 2020, and as a result the majority of the small proportion of respondents from July were reported outside of actual lockdown measures. However, changes to PA may persist, and the CCRR prospective cohort study is ongoing, with follow-up questionnaires sent to participants at regular intervals. When complete, this will allow for long-term impacts to be measured, accounting for seasonal variation and changes to restriction measures over time.

## CONCLUSIONS

Findings from our CCRR study suggest a significant decline in average PA levels in older adults following the introduction of lockdown measures during the COVID-19 pandemic. These are in keeping with similar decreases across age ranges, including healthy adults, children and adolescents, and in those with medical conditions,[12] and are particularly concerning given the negative health connotations of physical inactivity. Moreover, even before

the pandemic, older adults were more physically inactive than younger individuals, with only 61% and 40% of those aged 55–74 and 75 years old, respectively, meeting recommended levels of PA.[2]

In our study, lower activity levels after the introduction of lockdown were strongly linked to older age, and to those with objective markers of social isolation, subjective feelings of loneliness and symptoms of depression. Strategies and targeted interventions to increase and sustain PA levels in older adults are needed to mitigate the adverse health impacts of COVID-19-related lockdowns and of social isolation in general. A recent systematic review suggested that digital behavioural change interventions can increase PA levels, and decrease sedentary time, in older adults, and this may be an area of future research for PA in the context of social isolation.[11 60] Although there can be no 'one size fits all' approach,[13] interventions should consider social relationships in their design and implementation.

**Author affiliations**
[1]Department of Primary Care and Public Health, Imperial College London, London, UK
[2]MSk Lab, Department of Surgery and Cancer, Faculty of Medicine, Imperial College London, London, UK
[3]Ageing Epidemiology Research Unit (AGE), Faculty of Medicine, School of Public Health, Imperial College London, London, UK
[4]Public Health Directorate, Imperial College Healthcare NHS Trust, London, UK

**Acknowledgements** We are grateful to Lesley Williamson, Monica Munoz-Troncoso, Snehal Pandya and Emily Pickering (CHARIOT register and facilitator team); Mariam Jiwani, Rachel Veeravalli, Islam Saiful, Danielle Rose, Susie Gold, Rachel Nejade and Shehla Shamsuddin (Imperial College London student volunteers); Stefan McGinn-Summers, Neil Beckford, Inthushaa Indrakumar and Kristina Lakey (departmental administrative staff in AGE); Dinithi Perera (departmental manager); Heather McLellan-Young (project manager); Helen Ward, James McKeand, Geraint Price, Josip Car, Christina Atchison, Nicholas Peters, Aldo Faisal and Jennifer Quint (investigator team contributing to CCRR survey design, development and improvement).

**Contributors** DS, TB and CR conceived the paper, developed the survey materials, carried out the analysis, wrote the paper equally as joint lead authors and are the guarantors. CAdJL, PG, CTU-M and SA-A developed the survey materials, managed the cohort and data set and contributed to the analysis and writing and editing of the paper. AM, LTM and AHM developed the survey materials, supervised and managed the survey collection and analysis, and contributed to the writing and editing of the paper. All authors developed the survey, carried out the analysis and contributed to the development and editing of the paper.

**Funding** Work towards this article was in part supported by the National Institute for Health Research (NIHR) Applied Research Collaboration Northwest London and Imperial Biomedical Research Centre (BRC). DS and TB are supported by NIHR academic clinical fellowships.

**Disclaimer** The views expressed in this publication are those of the authors and not necessarily those of the National Institute for Health Research or the Department of Health and Social Care. Imperial College London is the sponsor for the CCRR study and has no influence on the direction or content of the work.

**Competing interests** LTM reports research funding from Janssen, Novartis, Merck and Takeda, outside the submitted work.

**Patient and public involvement statement** Older adult volunteers (60–80 years of age) from various social and cultural backgrounds provided feedback on the survey content. This feedback was incorporated into the survey design.

**Patient consent for publication** Not required.

**Ethics approval** This research was approved by the Imperial College Research and Ethics Committee (ICREC) and Joint Research Compliance Office (22/04/2020; 20IC5942). All participants were required to provide informed consent before taking part in the study. Data collected as a part of this study are anonymised and kept strictly confidential in accordance with the UK General Data Protection Regulations (2016).

**Provenance and peer review** Not commissioned; externally peer reviewed.

**Data availability statement** Data are available upon reasonable request. This is an ongoing study, but anonymised data can be provided upon request for the purposes of further data analysis, and can be requested from the data management coordinator, PG: parthenia.giannakopoulou13@imperial.ac.uk.

**ORCID iDs**
David Salman http://orcid.org/0000-0002-1481-8829
Celeste A de Jager Loots http://orcid.org/0000-0003-0789-3297

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
