## [Reviewer comments · BMJ Open]

ARTICLE DETAILS

TITLE (PROVISIONAL)	The impact of social restrictions during the COVID-19 pandemic on the physical activity levels of adults aged 50-92 years: a baseline survey of the CHARIOT COVID-19 Rapid Response prospective cohort study
AUTHORS	Salman, David; Beaney, Thomas; Robb, Catherine; de-Jaegar Loots, Celeste; Giannakopoulou, Parthenia; Udeh-Momoh, Chinedu; Ahmadi-Abhari, Sara; Majeed, Azeem; Middleton, Lefkos; McGregor, Alison

VERSION 1 – REVIEW

REVIEWER	Eva Maria Leon Zarceño Universidad Miguel Hernández
REVIEW RETURNED	19-Apr-2021

GENERAL COMMENTS	The article rigorously analyses the effect of isolation in the elderly population on different variables, being its analysis interesting and especially important the impact on physical activity. The main strength of the article is the large sample (N=6219) and the repeated measures survey. The limitations are mentioned by the authors themselves. The manuscript presents a clear and easily understandable language. The subject addressed is interesting and worrying from a social and health point of view, both in terms of the population at which the study is aimed and in terms of understanding the difficult time they are living through due to the pandemic. The Background and Rationale includes the essentials for understanding the objective of this study, although it would be advisable to complete it by expanding on the psychological characteristics that affect the elderly in general and thus justifying the reason for choosing this population. In this same section, it would be appropriate to expand it with other studies of changes in physical activity during COVID-19 confinement in a different age population. The method is correct. The instruments used are adequate for the evaluation of the variables, the IPAQ being one of the most widely used to measure physical activity and the HAD one of the most widely used instruments to evaluate depression and anxiety. However, I would suggest that the authors provide more information regarding the description of the instruments and the reliability indices of the questionnaires in this study and the response ranges (IPAQ, HADS and FRAIL). I would also suggest the authors to detail the inclusion and exclusion criteria. The hypotheses are correctly stated. Data analyses are rigorous and adequate to meet the objectives of the study. They are well structured and the presented analyses of predictors facilitate reporting. Recommendations to the authors: A minor correction: Pag 6. Line 55: GP - Indicate that this is the first
--

	time it appears in the text after the abbreviation. Background and Rationale: It would be advisable to complete it by expanding on the psychological characteristics that affect the elderly in general and thus justify the reason for choosing this population. In this same section it would be appropriate to expand it with other studies of changes in physical activity during COVID-19 confinement in a different age population. Indicate reliability data for the IPAQ, HADS and FRAIL in this sample. Describe the instruments and response ranges. I would suggest that the authors provide more information regarding the description of the instruments and the reliability indices of the questionnaires in this study and the response ranges (IPAQ, HADS and FRAIL). Explain inclusion and exclusion criteria.
--	--

REVIEWER	Francois Routhier Centre for Interdisciplinary Research in Rehabilitation and Social Integration, Institut de réadaptation en déficience physique de Québec, Québec City, Québec, Canada.
REVIEW RETURNED	19-Apr-2021

GENERAL COMMENTS	This paper is very well structured and clear. The subject is quite interesting. To make sure that all the journal's criteria are met, I would suggest adding information in the text regarding ethics and consent (already in the abstract). To facilitate comprehension, because it's a concept that is very important in your study, I would suggest including the paragraph explaining the MET calculation (from supplementary file 2) in the text. I would also like to know if the working status decreased the level of activity (were workers working from home or outside the home and did that influence the level of activity). It would be helpful to give a timeline of the different measures put in place (if there were any changes along the way). The decreased activity after lockdown in July might be easier to understand (temperature, new events/measures?), otherwise shouldn't all three months present similar results? Figure 4 is really hard to read, I would suggest increasing the font size. In the results section, you sometimes refer to figure 1 and 2, that use the entire sample, however you seem to want to justify changes between groups which we cannot see on these figures (i.e., demographic and lifestyle factors). I don't understand how the figure complements the provided data. Few suggestions are made to improve the situation or to lead to subsequent studies to improve PA in the evaluated group. Comparison with other age groups as well as in non pandemic contexts would be appreciated. It would be interesting to discuss why the cut off was July? Should we reevaluate at a later time to determine if the chronicity of the lockdown further decreases PA?
---

REVIEWER	Rozangela Verlengia Universidade Metodista de Piracicaba, Physical Education
REVIEW RETURNED	15-May-2021

GENERAL COMMENTS	Overall Comments: Thank you for the opportunity to review the manuscript entitled, "The impact of social restrictions during the COVID-19 pandemic on the physical activity levels of older adults: a baseline analysis of the CHARIOT COVID-19 Rapid Response prospective cohort study". This manuscript describes the finding of the baseline analysis of a prospective observational study with middle-aged and older adult participants in the North West London region investigating the
--

	impact of COVID-19 pandemic lockdown (from April to July 2020) on the PA level. This topic is part of an important area of the journal audience, providing information about the changes of physical activity level and associations factors using a validated questionnaire (IPAQ short version). The central negative aspect of the study is related to the self-reported PA overestimation. As highlighted by the authors in the first paragraph of the introduction, it is already well established that a large part of the older adult population does not contemplate the minimum MVPA recommendations (600 METs min/week). However, the study results showed median values of 2,772 and 2,440 METs min/week before and during the lockdown (highly active level); > 90% of the participants were classified as physically active in both periods. Although the authors report a significant reduction in MVPA after introducing lockdown in the UK, most of the participants remained physically active (> 600 METs min/week). The authors must describe these results more precisely and indicate in greater depth the study discussion. Specific Comments: 1) Considering that the study participants reported being between 50 and 92 years old, it is suggested that the term “middle-aged” should be added to the title (The impact of social restrictions during the COVID-19 pandemic on the physical activity levels of middle-aged and older adults: a baseline analysis of the CHARLOT COVID-19 Rapid Response prospective cohort study”). 2) In some parts of the manuscript it is reported “after the lockdown” (for example, in abstract, lines 24-25: “Main outcome measures: Self-reported PA before and after lockdown”), however, by reading the article, one understands that the text refers to the period during the implementation of the lockdown. Suggestion, standardize the writing throughout the article by placing “during the lockdown”. 3) In the final adjusted analysis, there are several significant association values. Why did the author only choose some specific description in the abstract (BMI, alcohol, and diet, for example, are missing)? 4) Please indicate the values as negative (for example, -240 [95% CI: -120 to -360]) according to the study results (abstract). 5) Please revise the values of lines 31 (page 4, abstract) and 44 (page 8, results) – 3,185 and 3,186 METs min/week. 6) Suggestion to be added to the methods: inclusion and exclusion criteria adopted. 7) What is the criterion for defining “cognitively healthy adult”? 8) In the study it is reported that the IPAQ was used to assess physical activity and sitting time. However, the issue of bouts \geq 10 minutes of moderate to vigorous physical activity is not reported. Considering that the UK guidelines, as well as WHO guidelines (2020) no longer require the need for the accumulation of moderate to vigorous physical activity in bouts \geq 10 minutes, the lack of this information can confuse readers, since the values MET x min/week of physical activity were extremely high.
--	---

	9) In the methods sections, the authors must describe in more detail what were the lockdown restriction measures adopted by the local government during the study period (April to July). 10) Please add the information contained in the item "Ethics approval" (page 2, lines 17-26) also to the methods. 11) In table 1 the information about the "loneliness" variable is missing. 12) There are some inconsistencies in the missing values (table with participant characteristics) and the number of observations (table 3) used in the analysis, the authors should better clarify these aspects. For example, in table participant characteristics, there is no record of missing values for employment status (6,219 records). However, supplementary table 3 indicates 5,958 observations used in the analysis. 13) According to item 13 of the STROBE the stretch "The survey was sent to 15,000 CHARIOT participants via email, with a subsequent 25,000 contacted by post. 7,320 participants responded and completed the survey. Of these respondents, 6,219 completed IPAQ data both before and after introduction of lockdown measures and were included in the final analysis." (page 7; line 23-30), must be placed in the results section. 14) Please describe how many outliers were observed and excluded by the IPAQ questionnaire. 15) Add the limitations in the study: data is not collected face to face. This aspect could influence the reported values of physical activity and sitting time. On the other hand, the restriction imposed by the COVID-19 pandemic made it impossible to adopt such strategy for data collection. 16) On page 13 is reported "First, the CRR cohort appear more physically active than the general population. 90% of participants in CRR achieved minimum UK4 and WHO3 guidance, both before and following lockdown". Although they are similar in terms of volume and intensity of physical activity, UK and WHO (2010) guidelines present a difference that decisively impacts the volume of reported physical activity. WHO (2010) guideline requires activities of moderate to vigorous intensity to be performance in sessions ≥ 10 minutes (which is in line with the IPAQ), while UK guideline (and the most current version of the WHO physical activity guideline, published in November 2020) eliminates the need to perform activities in bouts ≥ 10 minutes. As the physical activity was evaluated by the IPAQ, it is understood that the correct thing is to report that 90% of the studied sample complied with the WHO guideline (2010). So, it is necessary to adjust the sentence. 17) In addition, also on page 13, line 49-50 is reported "in contrast, only 61% of UK adults aged 55-74 years achieve minimum recommended levels. 2". Does this information refer to compliance with the WHO (2010) or UK guideline? Considering the physical activity data of moderate to vigorous intensity were collected based on bouts ≥ 10 minutes (as stated in the IPAQ), if reference 2 concerns the UK guideline, the sentence should be removed or another reference that uses the WHO guideline (2010) as a parameter should be found, based on self-report measures.
--	--

18) Differentiate tables and figures from supplementary material, for example: Table S1 and figure S1.
--

VERSION 1 – AUTHOR RESPONSE

Reviewer: 1

Dr. Eva Maria Leon Zarceño, Universidad Miguel Hernández

Comments to the Author:

The article rigorously analyses the effect of isolation in the elderly population on different variables, being its analysis interesting and especially important the impact on physical activity. The main strength of the article is the large sample (N=6219) and the repeated measures survey. The limitations are mentioned by the authors themselves. The manuscript presents a clear and easily understandable language. The subject addressed is interesting and worrying from a social and health point of view, both in terms of the population at which the study is aimed and in terms of understanding the difficult time they are living through due to the pandemic. The Background and Rationale includes the essentials for understanding the objective of this study, although it would be advisable to complete it by expanding on the psychological characteristics that affect the elderly in general and thus justifying the reason for choosing this population. In this same section, it would be appropriate to expand it with other studies of changes in physical activity during COVID-19 confinement in a different age population. The method is correct. The instruments used are adequate for the evaluation of the variables, the IPAQ being one of the most widely used to measure physical activity and the HAD one of the most widely used instruments to evaluate depression and anxiety. However, I would suggest that the authors provide more information regarding the description of the instruments and the reliability indices of the questionnaires in this study and the response ranges (IPAQ, HADS and FRAIL). I would also suggest the authors to detail the inclusion and exclusion criteria. The hypotheses are correctly stated. Data analyses are rigorous and adequate to meet the objectives of the study. They are well structured and the presented analyses of predictors facilitate reporting.

Thank you for these constructive and helpful comments. We have attempted to respond to the points raised as below:

Recommendations to the authors:

A minor correction: Pag 6. Line 55: GP - Indicate that this is the first time it appears in the text after the abbreviation.

Thank you – this has been amended

Background and Rationale: It would be advisable to complete it by expanding on the psychological characteristics that affect the elderly in general and thus justify the reason for choosing this population. In this same section it would be appropriate to expand it with other studies of changes in physical activity during COVID-19 confinement in a different age population.

Thank you – we have expanded this section to include the relevance of PA in older adults in particular, and how studies to date have focussed on younger adults and children in section 1.0 Physical activity (PA) is important in the prevention of sarcopenia, frailty and decreased functional ability in older adults.¹¹ Data collected on the pandemic, predominantly in younger adults and children, suggests a decrease in PA and an increase in sedentary time.¹² Given the increased susceptibility to physical inactivity and social isolation in older adults in particular, this is an important area of study.¹³

Indicate reliability data for the IPAQ, HADS and FRAIL in this sample. Describe the instruments and response ranges.

I would suggest that the authors provide more information regarding the description of the instruments and the reliability indices of the questionnaires in this study and the response ranges (IPAQ, HADS and FRAIL).

We have added reliability and response data as recommended in section 2.1, paragraph 2. Please also see the full questionnaire in supplementary data file 2.

For physical activity, the International Physical Activity Questionnaire (IPAQ) short-form (last 7 days) was used,¹⁴ asking respondents to document their weekly vigorous and moderate activity, walking and sitting time from the week prior to completing the survey; and for the week prior to implementation of social restriction measures. This has test-retest reliability of 0.75 in those under the age of 60

years.¹⁵ However, although less commonly studied in older populations, one study demonstrated reduced reliability, at 0.65 and 0.57 for men and women respectively aged 65-74 years, and 0.50 and 0.56 for those aged 75-89 years.¹⁶ For assessing frailty, the 5-point FRAIL scale,^{17,18} (ordinal scale 1-5; predictive validity for mortality up to 10 years; HR: 2.60)¹⁹ and for assessing mental health symptoms, the Hospital Anxiety and Depression (HADS) scale;²⁰ sensitivity and specificity 0.8 for both anxiety and depression;²¹ 14 questions on feelings related to anxiety and depression rated on a 4-level Likert scale) were used.

Explain inclusion and exclusion criteria.

Thank you – we have added more detail here to as suggested into section 2.1, paragraph 2:

“Participants were eligible for recruitment if they were participating in the CHARIOT Register, or were a consenting member of the household who wished to take part; had mental capacity to consent to participate; were willing and able to undertake an electronic questionnaire survey; were able to read, write and were fluent in English or identify an informant who was able to. Participants were excluded where they were no longer participating in the CHARIOT register, or if they did not have access to electronic devices to complete the questionnaire surveys.”

Reviewer: 2

Dr. Francois Routhier, Centre for Interdisciplinary Research in Rehabilitation and Social Integration, Institut de réadaptation en déficience physique de Québec, Québec City, Québec, Canada.

Comments to the Author:

This paper is very well structured and clear. The subject is quite interesting.

Thank you for your positive comments on our paper.

To make sure that all the journal's criteria are met, I would suggest adding information in the text regarding ethics and consent (already in the abstract).

Done as suggested added to the end of methods section 2.1

To facilitate comprehension, because it's a concept that is very important in your study, I would suggest including the paragraph explaining the MET calculation (from supplementary file 1) in the text.

Thank you – added to section 2.2:

Metabolic Equivalent of Task (MET) minutes per week, calculated for each activity and total activity (where 3.3 METS is considered equivalent to walking, and moderate and vigorous activity equivalent to 4 and 8 METS, respectively). To calculate the continuous variable of total MET minutes per a week, the self-reported duration (minutes) and frequency (days) of each of these PA categories is multiplied by the by the specified metric - supplementary file 1

I would also like to know if the working status decreased the level of activity (were workers working from home or outside the home and did that influence the level of activity).

Many thanks for this useful comment. 70% of participants were retired. Of those working, the majority were working from home, and we feel that the numbers here are too small to attempt to draw any inferences, although we agree that this is an important area of interest for further work.

It would be helpful to give a timeline of the different measures put in place (if there were any changes along the way).

Thank you, and we agree this would be useful to contextualise the findings. We have added this as a figure in supplementary file 1 (supplementary file 1: figure S5) and referenced this in the text.

The decreased activity after lockdown in July might be easier to understand (temperature, new events/measures?), otherwise shouldn't all three months present similar results?

Our aim in the analysis was to identify difference in activity after the introduction of lockdown measures in March 2020. We anticipated that any changes might persist even after the relaxation of social restriction measures. For this reason, the study data collection continued throughout July, despite a relaxation in social restrictions. Temperature, rainfall and seasonal effects will contribute to

physical activity. While we have not included these explicitly in the analysis, we have adjusted for monthly differences, which will pick up some of the confounding from environmental factors.

Figure 4 is really hard to read, I would suggest increasing the font size.

Thank you for picking up on this – we have made supplementary file 1: figure S4 landscape and increased its size.

In the results section, you sometimes refer to figure 1 and 2, that use the entire sample, however you seem to want to justify changes between groups which we cannot see on these figures (i.e., demographic and lifestyle factors). I don't understand how the figure complements the provided data. Thank you for pointing this out – this is because we had referenced both figures 1 and 2 at the end of a section, but agree this could be misinterpreted. We have therefore referenced the figures separately where appropriate for clarity and to avoid confusion – section 3.0.

Few suggestions are made to improve the situation or to lead to subsequent studies to improve PA in the evaluated group.

Thank you: we have added some comments regarding this in section 4.7 regarding recent evidence on digital interventions.

Comparison with other age groups as well as in non pandemic contexts would be appreciated.

Thank you: we have added some comments regarding this in section 4.7.

It would be interesting to discuss why the cut off was July?

As noted above, social restrictions were relaxed in July, and so a decision was made to stop further baseline data collection, but to invite follow-up surveys for those who had already completed a baseline survey. These follow-up surveys are on-going, and will be re-evaluated once all survey rounds are completed.

Should we re-evaluate at a later time to determine if the chronicity of the lockdown further decreases PA?

Thank you for this suggestion, we certainly plan to re-evaluate again once the CCRR study has come to an end, and further cycles of responses have been collected. This would allow for further seasonal effects to be incorporated, as you have suggested.

Reviewer: 3

Dr. Rozangela Verlengia, Universidade Metodista de Piracicaba Comments to the Author:

Overall and specific comments attached.

Overall Comments:

Thank you for the opportunity to review the manuscript entitled, "The impact of social restrictions during the COVID-19 pandemic on the physical activity levels of older adults: a baseline analysis of the CHARIOT COVID-19 Rapid Response prospective cohort study". This manuscript describes the findings of the baseline analysis of a prospective observational study with middle-aged and older adult participants in the North West London region investigating the impact of COVID-19 pandemic lockdown (from April to July 2020) on the PA level. This topic is part of an important area of the journal audience, providing information about the changes of physical activity level and associations factors using a validated questionnaire (IPAQ short version).

The central negative aspect of the study is related to the self-reported PA overestimation. As highlighted by the authors in the first paragraph of the introduction, it is already well established that a large part of the older adult population does not contemplate the minimum MVPA recommendations (600 METs min/week). However, the study results showed median values of 2,772 and 2,440 METs min/week before and during the lockdown (highly active level); > 90% of the participants were classified as physically active in both periods.

Although the authors report a significant reduction in MVPA after introducing lockdown in the UK, most of the participants remained physically active (> 600 METs min/week). The authors must describe these results more precisely and indicate in greater depth the study discussion.

Thank you for your thorough review and constructive comments for our article. We agree that PA levels are higher across our cohort than expected and have discussed this as a key limitation. We have addressed each other point in turn below.

Specific Comments:

1) Considering that the study participants reported being between 50 and 92 years old, it is suggested that the term “middle-aged” should be added to the title (The impact of social restrictions during the COVID-19 pandemic on the physical activity levels of **middle-aged** and older adults: a baseline analysis of the CHARIOT COVID-19 Rapid Response prospective cohort study”).

Thank you: we have changed the title to simply present the age range for accuracy, and to prevent the use of these terms

2) In some parts of the manuscript it is reported “after the lockdown” (for example, in abstract, lines 24-25: “Main outcome measures: Self-reported PA before and after lockdown”), however, by reading the article, one understands that the text refers to the period during the implementation of the lockdown. Suggestion, standardize the writing throughout the article by placing “during the lockdown”.

As raised by one of the other reviewers, we have now included a timeline of the social restrictions from March onwards in the UK (supplementary file 1: figure S5). As seen, social restrictions are dynamic over time, and are not uniform. Baseline data collection continued until 22nd July, and we decided to include these responses, as they are still relevant for documenting the change in PA that has occurred following the introduction of lockdown measures. We may expect reductions in physical activity to persist beyond the end of social restrictions, particularly in the context of an ongoing pandemic, where more vulnerable adults continue to avoid social contact. For these reasons, presenting this as ‘after the introduction of lockdown’ is more accurate and we have amended this where relevant.

We have added text to the limitations section of discussion addressing these points.

3) In the final adjusted analysis, there are several significant association values. Why did the author only choose some specific description in the abstract (BMI, alcohol, and diet, for example, are missing)?

Thank you, we would have liked to include these measures in the abstract but were unable due to the word count and so have focussed on those with stronger effect sizes/more novel findings

4) Please indicate the values as negative (for example, -240 [95% CI: -120 to -360]) according to the study results (abstract).

We have written in the abstract that values are ‘less’ in these groups. We feel this makes the description applied in the abstract clearer than were we to use negative sign, as this could otherwise mistakenly be interpreted as a double negative which might confuse readers. In the figures, we have used minus signs. We are happy to change in the abstract if the editors feel it would be clearer.

5) Please revise the values of lines 31 (page 4, abstract) and 44 (page 8, results) – 3,185 and 3,186 METs min/week.

Thank you for spotting this. The 3,186 in results represents a rounding error which has been corrected to 3,185.

6) Suggestion to be added to the methods: inclusion and exclusion criteria adopted.

Thank you – this has been done

7) What is the criterion for defining “cognitively healthy adult”?

Thank you for highlighting this. We have changed the text to make this more clear in section 2.1 to indicate that these are individuals without a known diagnosis of dementia.

8) In the study it is reported that the IPAQ was used to assess physical activity and sitting time. However, the issue of bouts ≥ 10 minutes of moderate to vigorous physical activity is not reported. Considering that the UK guidelines, as well as WHO guidelines (2020) no longer

require the need for the accumulation of moderate to vigorous physical activity in bouts ≥ 10 minutes, the lack of this information can confuse readers, since the values MET x min/week of physical activity were extremely high.

This is an important point; thank you for raising it. Although the UK CMO guidance changed regarding this point in 2019, and the WHO guidance in November 2020 we have clarified this in the methods, explaining that periods <10 minutes were excluded for consistency with IPAQ cleaning protocol. We highlight in the methods the numbers excluded, which make apparent that the numbers are relatively low (the highest number of excluded entries was 25, for reported vigorous activity, during lockdown) and unlikely to impact the results in this case. We have added to the methods section: "Periods of activity under 10 minutes were excluded as per the protocol, excluding for vigorous, moderate and walking activities during lockdown, 25, 23 and 12 periods, respectively (for pre-lockdown activity, excluding 10, 13 and 3 periods of activity, respectively)."

9) In the methods sections, the authors must describe in more detail what were the lockdown restriction measures adopted by the local government during the study period (April to July). We agree that this would be useful, and have now added a timeline to supplementary file 1, showing the changes to social restrictions over time.

10) Please add the information contained in the item "Ethics approval" (page 2, lines 17-26) also to the methods.
This has been done

11) In table 1 the information about the "loneliness" variable is missing.
Thank you for spotting this omission. We have updated Table 1 with the loneliness variable.

12) There are some inconsistencies in the missing values (table with participant characteristics and the number of observations (table 3) used in the analysis, the authors should better clarify these aspects. For example, in table participant characteristics, there is no record of missing values for employment status (6,219 records). However, supplementary table 3 indicates 5,958 observations used in the analysis.

Thank you for raising this. Table 1 presents the results of the survey questions as asked. In order to include in regression analyses, we categorised responses for employment into four groups (employed, furloughed, unemployed, retired). Those specified as 'none of the above' were set to missing, explaining the discrepancy. We have added text to the methods to explain that employment categories were re-categorised, and referenced further detail in the supplementary appendix.

Table 3 additionally presents multivariable analyses. As documented in Table 1, age, sex and ethnicity are not fully complete, and hence for multivariable analyses, the complete case analysis will not be on the full set of 6,219 participants. This is referenced in the methods: "Denominators for each model vary according to the levels of missingness in variables included in the models, which was low for most variables, except for BMI (unrecorded in 51.4% of participants)."

13) According to item 13 of the STROBE the stretch "The survey was sent to 15,000 CHARLOT participants via email, with a subsequent 25,000 contacted by post. 7,320 participants responded and completed the survey. Of these respondents, 6,219 completed IPAQ data both before and after introduction of lockdown measures and were included in the final analysis." (page 7; line 23-30), must be placed in the results section.
Thanks for this advice, we have repositioned the text to the first paragraph of results.

14) Please describe how many outliers were observed and excluded by the IPAQ questionnaire. Thank you, we have added a comment to the methods explaining that periods under 10 minutes were removed, as recommended, and also in results, reporting the numbers across each of the types of activity, before and after lockdown:
"Periods of activity under 10 minutes were excluded as per the protocol, excluding for vigorous, moderate and walking activities during lockdown, 25, 23 and 12 periods, respectively (for pre-lockdown activity, excluding 10, 13 and 3 periods, respectively)."

15) Add the limitations in the study: data is not collected face to face. This aspect could influence the reported values of physical activity and sitting time. On the other hand, the restriction imposed by the COVID-19 pandemic made it impossible to adopt such strategy for data collection.

Thank you for raising this valuable point: we have added this comment to the limitations section 4.6: Finally, recall bias and seasonal changes in physical activity may also have impacted on the results, with the additional factor that data were collected remotely rather than face to face (although there was necessary due to pandemic control measures).

16) On page 13 is reported “First, the CCRR cohort appear more physically active than the general population. 90% of participants in CCRR achieved minimum UK4 and WHO3 guidance, both before and following lockdown”. Although they are similar in terms of volume and intensity of physical activity, UK and WHO (2010) guidelines present a difference that decisively impacts the volume of reported physical activity. WHO (2010) guideline requires activities of moderate to vigorous intensity to be performance in sessions ≥ 10 minutes (which is in line with the IPAQ), while UK guideline (and the most current version of the WHO physical activity guideline, published in November 2020) eliminates the need to perform activities in bouts ≥ 10 minutes. As the physical activity was evaluated by the IPAQ, it is understood that the correct thing is to report that 90% of the studied sample complied with the WHO guideline (2010). So, it is necessary to adjust the sentence.

Thank you, this sentence has been adjusted as suggested.

17) In addition, also on page 13, line 49-50 is reported “in contrast, only 61% of UK adults aged 55-74 years achieve minimum recommended levels. 2”. Does this information refer to compliance with the WHO (2010) or UK guideline? Considering the physical activity data of moderate to vigorous intensity were collected based on bouts ≥ 10 minutes (as stated in the IPAQ), if reference 2 concerns the UK guideline, the sentence should be removed or another reference that uses the WHO guideline (2010) as a parameter should be found, based on selfreport measures.

Thank you; this figure is from the UK Active Lives Survey 2018-2019 which only included activity in bouts of 10 minutes or more, which is consistent with WHO 2010 guidance. We have therefore ensured that this sentence references the WHO 2010 guideline.

18) Differentiate tables and figures from supplementary material, for example: Table S1 and figure S1.

Thank you, this has been done as suggested

VERSION 2 – REVIEW

REVIEWER	Francois Routhier Centre for Interdisciplinary Research in Rehabilitation and Social Integration, Institut de réadaptation en déficience physique de Québec, Québec City, Québec, Canada.
REVIEW RETURNED	03-Aug-2021
GENERAL COMMENTS	The authors have adequately addressed all my comments.
REVIEWER	Rozangela Verlengia Universidade Metodista de Piracicaba, Physical Education
REVIEW RETURNED	29-Jul-2021
GENERAL COMMENTS	The researchers complied with the requests. So, I recommend accepting the article for publication.